# Rethinking $SO(3)$-equivariance with Bilinear Tensor Networks

## Abstract

Many datasets in scientific and engineering applications are comprised of objects which have specific geometric structure. A common example is data inhabiting a representation of $SO(3)$ scalars, vectors, and tensors. One way to exploit prior knowledge of the structured data is to enforce $SO(3)$-equivariance. While general methods for handling arbitrary $SO(3)$ representations exist, they can be computationally intensive and complicated to implement. We show that by judicious symmetry breaking, we can efficiently increase the expressiveness of a network operating on these representations. We demonstrate the method on an important classification problem from High Energy Physics known as *b-tagging*. In this task, we find that our method achieves a $2.7\times$ improvement in rejection score over standard methods.

## 1 Introduction

In many Machine Learning (ML) applications, at least some of the data of interest have specific geometric structure. For example, position measurements from LiDAR imaging, the configuration of atoms in molecular potentials, and measurements of particle momenta are all cases where the data are naturally represented as spatial 3-vectors. However, classical Neural Network (NN) architectures are not well suited to this sort of data; for instance, the standard Multi Level Perceptron would require that all information, spatial or otherwise, must be collapsed into a flat list of features as input to the network. In this case, the spatial nature of the data, while not lost, is not communicated *a priori* nor enforced *post hoc*.

More recently, developments in the field of Representation Learning have shown that *equivariant* NNs are a natural way to accommodate structured data, and in many cases lead to substantially improved algorithms. Very informally, a function (such as a NN) is called equivariant if the output transforms similarly to the input.

Convolutional Neural Networks (CNNs) are the prototypical example of this. CNNs exploit the fact that image data can be most naturally represented as data on a discrete 2-dimensional grid. This data structure is associated with the representation of the group of discrete translations. The standard CNN layer takes advantage of this by operating on input grid (pixel) data with discrete translation operations, and returning outputs on a similar grid structure. Because the output of each layer has the same representational structure as the input, it is straightforward to build very deep representations without destroying the prior spatial structure of the data, simply by stacking CNN layers. The result, of course, is that CNNs have completely revolutionized the field of computer vision.

We specifically consider the case of continuous scalar and 3-dimensional vector point data, as may be encountered in many point-cloud datasets. For these data, the natural group associated with their representation is $SO(3)$, the set of 3D rotations. Therefore, one strategy to incorporate this structure

Submitted to 37th Conference on Neural Information Processing Systems (NeurIPS 2023). Do not distribute.

into a neural architecture is to enforce equivariance *w.r.t.* SO(3), and several such architectures have been proposed [1, 2, 3]. In general, these approaches achieve equivariance either by defining a spherical convolutional operation [3, 1], or by constraining the network's operations to maintain strict representational structure [2, 4].

Our method follows the latter approach, but in a much simpler way. Rather than concerning ourselves with arbitrary $(2\ell + 1)$-dimensional representations, we consider only a few physically relevant representations: scalars, vectors, and order-2 tensors. For these three representations, it is straightforward to enumerate the options for linear neuron layers. We also want our network to be able to exchange information between different representations. The Clebsh-Gordon theory prescribed in other methods provides the most general method for projecting arbitrary tensor products between representations back into irreducible representations, However, once again we take a similar approach, and instead introduce a simple *Tensor Bilinear Layer*, a subset of the CG space that consists of commonly known and physically intuitive operations, such as the vector dot product and cross product.

Importantly, we propose a novel method that allows us to relax equivariance requirements when an axial symmetry is present, by allowing the global SO(3) symmetry to be locally broken down to SO(2). These looser conditions allow us to design of models that enforce only the instantaneously relevant equivariance, and allows the network to learn more expressive functions at each layer. We show that this kind of equivariant neuron is generally only possible with the introduction of order-2 tensor representations, but we provide an efficient implementation for vector-valued networks that constructs only the minimal tensors required.

To illustrate a real-world application to data with an axial symmetry, we introduce a common problem from the field of High Energy Physics (HEP), described in Sec. 2. In Sec. 3, we describe the modular elements of our method, from which a wide variety of neural architectures may be composed. In Sec. 4, we describe a specific architecture based on Deep Sets [5] which will serve as a baseline model, and we illustrate how to adapt this architecture using our approach. In Sec. 5, we describe the simulated data used for training and evaluation, and describe the results of a progressive implementation of the modules developed herein. Finally, we offer concluding remarks in Sec. 6.

## 1.1 Related Work

From the field of High Energy Physics, there has been much work in applying various DL approaches to jet tagging [6, 7, 8, 9] in general and b-tagging in particular [10, 11]. The present work seeks to build on this effort by offering novel neural architectures that can be adapted into next-generation applications.

From the field of Machine Learning, there have been numerous prior works on SO(3) equivariant models [1, 2, 3, 4]. In general, these approaches depend on Clebsh-Gordon (CG) decomposition and/or sampling from spherical harmonics. While our approach is more similar to the CG method, it is simpler and more relevant for the task at hand. Moreover, we innovate on the the space of allowed equivariant operations by relaxing the global SO(3) symmetry which is relevant for our particular application.

## 1.2 Novel Developments

The main innovation of this paper is to expand the set of linear equivariant maps in the special case where there is a "special" direction in space, which may change from sample to sample. In this case, it is possible to maintain global SO(3) equivariance, while breaking the per-layer equivariance condition down to a locally-defined SO(2) symmetry, which is parameterized by the special direction. We also innovate by introducing a simpler method of forming SO(3)-equivariant nonlinearities, by simply introducing familiar bilinear operations on spatial representations such as scalars, vectors, and tensors. In addition to the nonlinearity provided by the bilinear operations, we also introduce simple nonlinear activation functions on the vector and tensor representations, which we find helps stabilize training and improve performance.

Lastly, from the physics perspective, we propose a significant departure from standard practice, by stipulating that our b-tagging should be provided with raw 3-dimension position and momentum information, as this is the only way to ensure that SO(3)/SO(2) equivariance is exploited.

While we demonstrate our methods using a specifc architecture based on Deep Sets [5], we expect these innovations can be useful in many other applications. Given the modularity and strictly defined input and output representations of each layer, these elements could be used to augment other neural architectures such as convolutional, graph, and transformers as well.

## 2 B-jet Identification at LHC Experiments

In HEP experiments, such as ATLAS [12] and CMS [13] at CERN, *b-jets* are a crucial signal for studying rare phenomena and precision physics at the smallest scales of nature. A *jet* is a collimated spray of hadronic particles originating from energetic quarks or gluons produced in high energy particle collisions. A *b-jet* is a jet which specifically originates from a $b$-quark; when these quarks hadronize, they form metastable B-mesons which travel some distance from the collision origin before decaying, emitting particles from a secondary, displaced vertex.

Charged particles originating from these vertices are measured with tracking detectors and are often referred to as *tracks*. Due to the displacement of the secondary vertex, when track trajectories originating from B-meson decays are extrapolated backwards, they are generally not incident to the origin. Therefore, we instead measure the distance to the point of closest approach; this is often referred to as the *track impact parameter*, which is a 3-vector quantity that we denote with $\mathbf{a}$.

In most applications, only the transverse and longitudinal components, $d_0$ and $z_0$, of this impact parameter are examined [14]. The magnitude of these projections is the most distinctive feature that indicates whether a particular jet originated from a $b$-quark.

The inspiration for this work was the observation that the physical processes which govern how particles within a jet are produced and propagated are largely invariant with respect to rotations about the *jet axis*, denoted $\hat{\mathbf{j}}$. This is the unit vector in the direction of the aggregate jet's momentum vector. On the other hand the standard $b$-tagging observables $d_0$ and $z_0$ have no well-defined transformation rule under rotations, *i.e.* they are not part of a covariant representation.

Previous works [8] have demonstrated that networks which exploit this natural $\mathrm{SO}(2)$ symmetry can greatly improve performance, but these methods all essentially rely on reducing the problem to vectors in a 2-dimensional plane. In order to obtain an equivariant representation in the case of $b$-jets, we must consider the full 3-dimensional structure of the impact parameter, which transforms as a vector under general rotations $\mathbf{a} \xrightarrow{R} R\mathbf{a}$. In addition to the 3-dimensional impact parameter $\mathbf{a}$, we also have information about the track's momentum $\mathbf{p}$ and various scalar quantities such as the particle's charge, energy, and a limited identification of the particle type.

In the next section, we will describe modular neural elements that can solve this problem, by allowing a network to admit a global $\mathrm{SO}(3)$ symmetry which preserves the scalar and vector representations, while also breaking $\mathrm{SO}(3)$ down to the more physically appropriate $\mathrm{SO}(2)$ whenever possible.

## 3 Network Elements

Our proposed method depends on three modular elements, described in detail in the following subsections. The overall strategy begins by mirroring what has proved to work for NNs in general: we interleave simple linear (or affine) layers with nonlinear activation functions, in order to learn powerful models. For an equivariant network, we first need to identify a set of linear equivariant maps suitable for the symmetry at hand. In our case, we come up with two sets of such maps: a global $\mathrm{SO}(3)$-equivariant affine layer, and a locally $\mathrm{SO}(2)_{\hat{\mathbf{j}}}$-equivariant linear layer.

Since we also require our network to mix between its scalar, vector, and tensor representations, we introduce an equivariant *bilinear* layer. Lastly, we define $\mathrm{SO}(3)$ equivariant nonlinear activations for each output representation.

In Sec. 4, we demonstrate how to combine these elements into a complete neural architecture. This architecture is based on the Deep Sets [5] architecture suitable for variable-length, permutation-invariant data.

## 3.1 $\mathrm{SO}(2)_{\hat{\mathbf{j}}}$-equivariant Linear Layers

A well-known way to ensure equivariance *w.r.t.* any group is to broadcast the neural action across the representational indices of the data [15, 16]. That is, the neural weight matrix simply forms linear combinations of the features in their representation space. In general, it is helpful to add a bias term, but care must be taken to select one that preserves equivariance.

The simplest example of this is for a collection of $F$ scalar input features, $\{s_i\}$, mapping to a collection of $K$ output features. The scalar has no representational indices, so this simply amounts to the standard affine[1] network layer

$$y_i = W_{ij}s_j + b_i \tag{1}$$

where the learnable parameters $W_{ij}$ and $b_i$ are the neural weights and bias terms, respectively. In the vector case, we may generalize to

$$\mathbf{y}_i = W_{ij}\mathbf{v}_j\,;\quad \mathbf{b}_i = 0\,. \tag{2}$$

Note that the equivariance condition for vector-valued functions $f(R\mathbf{v}) = Rf(\mathbf{v})$ implies that $R\mathbf{b} = \mathbf{b}$ for arbitrary rotation $R$; hence, the bias vector must be zero. Finally, the analogous case for order-2 tensors is:

$$Y_i = W_{ij}T_j + B_i\,;\quad B_i = b_iI\,, \tag{3}$$

where again we have learnable scalar parameters $b_i$. In this case, the equivariance condition is $f(RTR^T) = Rf(T)R^T$, which implies that $RBR^T = B$, *i.e.* $B$ must commute with arbitrary $R$. Therefore, $B$ must be proportional to the identity tensor $I$.

The above neurons are purely isotropic in $\mathrm{SO}(3)$. However, as discussed in Sec. 1, for our problem we have prior knowledge that the distribution is symmetric about a specific axis. At worst, having only isotropic operations can over-regularize the network by imposing too much structure, and at best it might be harder for the network to spontaneously learn about the axial symmetry. We therefore consider the most general linear map is equivariant *w.r.t.* the axial symmetry. Since this is a lesser degree of symmetry, the network should have greater freedom in choosing linear maps.

### 3.1.1 Vector Case

Let $\hat{\mathbf{j}}$ be a unit vector (in our application, the jet's momenutm axis) which is instantaneously fixed per batch input. The rotations about this axis define a proper subgroup $S_{\hat{\mathbf{j}}} \subset \mathrm{SO}(3)$ where we identify $S_{\hat{\mathbf{j}}} \cong \mathrm{SO}(2)$. We therefore refer to this subgroup as $\mathrm{SO}(2)_{\hat{\mathbf{j}}} \subset \mathrm{SO}(3)$; the distinction being that $\mathrm{SO}(2)_{\hat{\mathbf{j}}}$ fixes a representation on $\mathbb{R}^3$ which depends on $\hat{\mathbf{j}}$.

The set of all linear $\mathrm{SO}(2)_{\hat{\mathbf{j}}}$-equivariant maps is exactly the set of matrices $A$ which commute with arbitrary $R_{\hat{\mathbf{j}}} \in \mathrm{SO}(2)_{\hat{\mathbf{j}}}$, which are of the form

$$A = (a\hat{\mathbf{j}}\hat{\mathbf{j}}^T + b(I - \hat{\mathbf{j}}\hat{\mathbf{j}}^T))R'_{\hat{\mathbf{j}}}(\phi)\,, \tag{4}$$

for arbitrary learnable parameters $\bar{\theta} = (a, b, \phi)$. The first two terms represent anisotropic scaling in the directions parallel and perpendicular to $\hat{\mathbf{j}}$, respectively. The third term represents any other arbitrary rotation about the $\hat{\mathbf{j}}$ axis, parameterized by a single angle $\phi$.

Because $A$ commutes with all $R_{\hat{\mathbf{j}}} \in \mathrm{SO}(2)_{\hat{\mathbf{j}}}$, the linear layer defined by

$$\mathbf{y}_i = A_{\bar{\theta}_{ij}}\mathbf{v}_j \tag{5}$$

is $\mathrm{SO}(2)_{\hat{\mathbf{j}}}$-equivariant, for arbitrary parameters $\bar{\theta}_{ij}$. The isotropic linear neuron of Eq. 1 corresponds to the special case $a_{ij} = b_{ij}$, $\phi_{ij} = 0$.

### 3.1.2 Tensor Case

In order for a tensor-valued linear map $L$ to be equivariant, we require that $L(R_{\hat{\mathbf{j}}}TR_{\hat{\mathbf{j}}}^T) = R_{\hat{\mathbf{j}}}(LT)R_{\hat{\mathbf{j}}}^T$. Note that in the case of full $\mathrm{SO}(3)$ equivariance, the only option is for $L$ to be proportional to the

---

[1]Also referred to as a Dense or Linear layer.

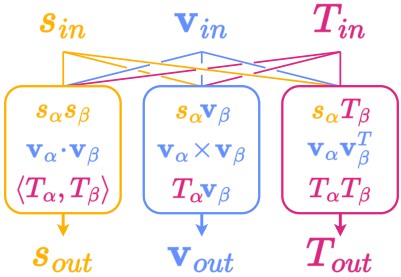

Figure 1: A schematic diagram of the bilinear layer with mixing between different representations.

identity. Without loss of generality, we may assume the order-4 tensor $L$ can be written as a sum
of terms $A \otimes B$ for some order-2 tensors $A, B$. The tensor product acts on an order-2 tensor $T$ as
$(A \otimes B)T := A T B^T$. Taking $L$ to be of this form (up to linear combinations), the equivariance
condition reads $A(R_{\hat{\mathbf{j}}} T R_{\hat{\mathbf{j}}}^T) B^T = R_{\hat{\mathbf{j}}}(A T B^T) R_{\hat{\mathbf{j}}}^T$. This is satisfied when both $A$ and $B$ commute
with $R_{\hat{\mathbf{j}}}$; we have already identified the set of such matrices in Eq. 4. Therefore, we define the action
of the tensor-valued $\mathrm{SO}(2)_{\hat{\mathbf{j}}}$ linear layer by:

$$Y_i = A_{\bar{\theta}_{ij}} T_j A_{\bar{\varphi}_{ij}}^T + b_i I \,, \tag{6}$$

where the parameters $(\bar{\theta}_{ij}, \bar{\varphi}_{ij})$ are the neural connections and we also allow for an affine bias term
parameterized by $b_i$, which is proportional to the identity tensor and hence also equivariant.

### 3.2 Tensor Bilinear Operations

So far we have provided two means for working with data in the $\mathrm{SO}(3)$ scalar, vector, and order-2
tensor representations. However, we also desire a means for allowing information between the
different representations to be combined and mixed.

The most general approach to this is addressed by Clebsh-Gordon theory [2, 4]. But we adopt a
simpler approach, wherein we take advantage of the familiar representations of our data and employ
common bilinear operations such as dot products and cross products for vectors[2]. This allows
the network to create a mixing between different representations. The operations considered are
enumerated schematically in Fig. 1. In order to form these terms, the bilinear layer requires that the
scalar, vector, and tensor inputs $(s, \mathbf{v}, T)$ all have the same size, $2F$, in their feature dimension, and
that the size is a multiple of two. We then split the features into groups of two: $s_a = \{s_i\}_{i=1..F}$,
$s_b = \{s_i\}_{i=F+1..2F}$, and define similarly $\mathbf{v}_{a,b}$ and $T_{a,b}$.

After effecting all of the options from Fig. 1, the layer returns scalar, vector, and tensor outputs with
$3F$ features each.

### 3.3 $\mathrm{SO}(3)$-equivariant Nonlinear Activations

For the scalar features, any function is automatically equivariant. Therefore, for these features we use
the well-known ReLU[17] activation function, although any alternative nonlinearity would also work.

In the vector and tensor cases, care must be taken to ensure equivariance. For the vector case, we
state a simple theorem[18]:

**Theorem 3.1** *For any vector-valued function $f : \mathbb{R}^3 \to \mathbb{R}^3$ which satisfies $f(R\mathbf{x}) = Rf(\mathbf{x})$ for all
$R \in \mathrm{SO}(3)$, there exists a scalar function $\tilde{f}$ such that*

$$f(\mathbf{x}) = \tilde{f}(|x|)\hat{\mathbf{x}} \,,$$

*where $\hat{\mathbf{x}} = \mathbf{x}/|x|$ when $|x| > 0$ and $\hat{\mathbf{x}} = \mathbf{0}$ otherwise.*

---

[2]Of course, these operations can be expressed in terms of the CG basis, but may not span the entire space of
irreducible representations guaranteed by Schur's lemma.

In other words, we may chose an arbitrary, nonlinear function $\tilde{f}$ which acts only on the vector magnitude, and the layer must leave the direction of the input unchanged. This leaves many possibilities; after some experimentation, we found the following activation, which we call Vector ReLU (VReLU), works well:

$$\mathrm{VReLU}(\mathbf{v}) := \begin{cases} \mathbf{v} & |v| < 1 \\ \mathbf{v}/|v| & \text{else} \end{cases} . \tag{7}$$

The VReLU activation is analogous to the standard rectified linear unit, except that the transition from linear to constant happens at a fixed positive magnitude rather than zero. We found that in particular, the saturating aspect of VReLU greatly helps to stabilize training, as otherwise the vector features tend to coherently grow in magnitude, leading to exploding gradients.

For the order-2 tensor case, we note here that the tensor analog to Theorem 3.1 is much more nuanced[18], and in general depends on three principal invariants $\mathcal{I}_1, \mathcal{I}_2, \mathcal{I}_3$. For simplicity, we define the Tensor ReLU (TReLU) completely analogously to the vector case, and leave a more complete analysis of tensor nonlinearities to future work:

$$\mathrm{TReLU}(T) := \begin{cases} T & ||T||_F < 1 \\ T/||T||_F & \text{else} \end{cases} . \tag{8}$$

# 4 Benchmark Architectures

We now have defined the four modular elements which provide the appropriate equivariant operations. In order to evaluate the practical effects of these modules, we define a benchmark architecture that is based on the Deep Sets architecture[5], also referred to as a Particle Flow Network (PFN) [19] in the field of HEP. The PFN is a commonly-used architecture for this sort of problem in real-world applications such as at the ATLAS experiment[14].

We will first define the standard PFN architecture, which will serve as our baseline in experiments. Then, we describe a modified version at the module level using the analogous equivariant operations in place of the standard neural network layers.

## 4.1 Particle Flow Network

The basic structure of the PFN [19] is based on the Deep Sets [5] architecture, and will serve as our baseline. It is of the form:

$$\mathrm{PFN}(\{p_k\}) = F\left(\sum_{k=1}^{P} \Phi(p_k)\right) . \tag{9}$$

where $\Phi : \mathbb{R}^F \to \mathbb{R}^L$ and $F : \mathbb{R}^L \to Y$ are arbitrary continuous functions parameterized by neural networks. $L$ is the dimension of the latent embedding space in which the particles are aggregated and $P$ is the number of particles in an observed jet. $Y$ represents the relevant output space for the task at hand; since our task is classification, we consider $Y = [0, 1]$.

The input features $\{p_k\}$ represent the observed track particles within the jet. These features include:

- The jet 3-momentum in detector coordinates, $(p_T^{(J)}, \eta^{(J)}, \phi^{(J)})$
- The 3-momentum of each particle track in *relative* detector coordinates, $(p_T^k, \Delta\eta^k, \Delta\phi^k)$
- The track impact parameters of each particle $(d_0^k, z_0^k)$
- The particle's charge $q$ and particle type {electron, muon, hadron}

For each jet, we allow up to $P = 30$ particle tracks; inputs with fewer than 30 particles are padded with zeros. We also repeat the jet 3-momentum over the particle axis and concatenate with the rest of the per-particle features. The discrete particle type feature is embedded into 3 dimensions. After concatenating all features, the input to the PFN is of shape $(*, P, F)$ where $F = 12$ is the feature dimension.

The subnetworks $\Phi$ and $F$ are simple fully-connected neural networks. $\Phi$ consists of two hidden layers with 128 units each, and ReLU activation. The output layer of $\Phi$ has $L$ units and no activation applied. The $F$ network consists of three hidden layers with 128 units each and ReLU activations.

The final output layer has two units with no activation, in order to train with a categorical cross entropy objective.

## 4.2 Vector and Tensor PFN

We now adapt the basic PFN architecture and promote it to what we term a Vector PFN (VPFN) or Tensor PFN (TPFN), according to the highest representation included. The overall architecture is of the same form as Eq. 9; we will simply modify the detailed implementation of the $\Phi$ and $F$ sub-networks.

The first change is that the input features now belong strictly to one of the three $\mathrm{SO}(3)$ representations: scalar, vector, or order-2 tensor:

$$\mathrm{TPFN}(\{(s, \mathbf{v}, T)_k\}) = F\left(\sum_{k=1}^{P} \Phi(s_k, \mathbf{v}_k, T_k)\right) \tag{10}$$

In general, the number of features in any of the representation channels are independent. The features for the TPFN experiments include:

- The jet 3-momentum in Cartesian coordinates $(p_x^{(J)}, p_y^{(J)}, p_z^{(J)})$
- The 3-momentum of each particle track $\mathbf{p}^k$
- The 3-position of the track's point of closest approach to the origin $\mathbf{a}^k$
- The charge and particle type of each track, as described in Sec. 4.1

As before, we replicate the jet momentum across the particle index, and we embed the particle type into 3 dimensions, resulting in $F_s = 4$ scalar and $F_v = 3$ vector features. Since there are no observed tensor features for particle tracks, we synthesize an initial set of features to act as a starting point for the tensor operations. This is done by taking the outer product between all combinations of the three available vector features, resulting in $F_t = 9$ features.

We now have $\Phi : \mathbb{R}^{F_s \times 3F_v \times 9F_t} \to \mathbb{R}^{L \times 3L \times 9L}$, where $F_s, F_v, F_t$ are the number of scalar, vector, and tensor inputs, respectively. A single layer of $\Phi$ is formed as shown in Fig. 2, by combining in sequence the Affine, $\mathrm{SO}(2)_{\mathbf{j}}$-Linear, Bilinear, and Nonlinear modules described in Sec. 3. The network consists of two hidden and one output layer. Each hidden Affine layer of the $\Phi$ network contains $2F = 128$ features per representation, which results in $3F = 192$ features after the Bilinear layer. The output of the $\Phi$ sub-network had $L$ features, and there is no Bilinear or Nonlinear layers applied.

The $F$ network is built similarly to the $\Phi$ network, except that it has three hidden tensor layers. In lieu of an output layer, after the hidden tensor layers, the $F$ network computes the square magnitude of each vector and tensor feature, in order to create a final set of $3 \times 3F$ scalar invariant features. The scalar features are concatenated, passed through two more hidden layers with 128 units each and ReLU activations, and finally to an output layer with two units and no activation.

## 5 Experiments

To train b-tagging algorithms, we must use Monte Carlo simulations of particle collision events, as this is the only way to get sufficiently accurate ground truth labels. The optimization of these algorithms is commonly studied by experiments such as ATLAS and CMS, which use highly detailed proprietary detector simulation software, and only limited amounts data are available for use outside of the collaborations. [20] There are also some community-generated datasets available for benchmarking [7], however none of these publicly-available datasets contain the key information that our method leverages for its unique equivariant approach. Specifically, our model requires the full 3-dimensional displacement vector of each track's impact parameter, whereas the existing datasets only retain the transverse and longitudinal projections $d_0$ and $z_0$. Therefore, we have created a new dataset for b-jet tagging benchmarks, to be made publicly available. The data is generated using standard Monte Carlo tools from the HEP community.

We begin by generating inclusive QCD and $t\bar{t}$ events for background and signal, respectively, using PYTHIA8[21]. PYTHIA handles sampling the matrix element of the hard processes at $\sqrt{s} = 13TeV$,

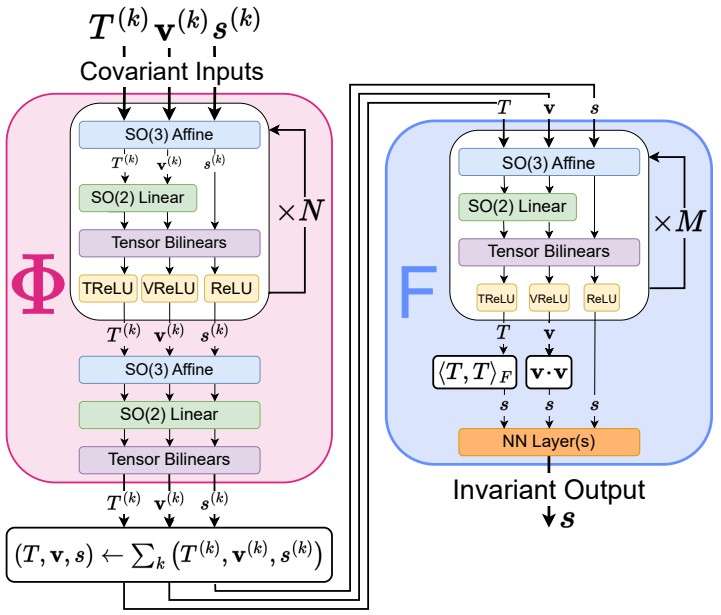

Figure 2: A schematic diagram of the DeepSets-adapated Tensor PFN.

the parton shower, and hadronization. The hadron-level particles are then passed DELPHES[22], a fast parametric detector simulator which is configured to emulate the CMS[13] detector at the LHC.

After detector simulation, jets are formed from reconstructed EFlow objects using the anti-$k_T$ [23, 24] clustering algorithm with radius parameter $R = 0.5$. Only jets with $p_T > 90$GeV are considered. For the signal sample, we additionally only consider jets which are truth-matched to a B-meson. Finally, the highest-$p_T$ jet is selected and the track and momentum features are saved to file.

The training dataset consists of a balanced mixture of signal and background with a total of 1M jets. The validation and test datasets contain 100k signal jets each. Due to the high degree of background rejection observed, we must generate a substantially larger sample of background events for accurate test metrics, so the validation and test datasets contain 300k background jets each.

## 5.1 Results

To quantify the performance of our model, we consider the following metrics in our experiments. First, the loss function used in the training is sparse categorical cross entropy, which is also used in the validation dataset. We also consider the area under the ROC curve (AUC) as an overall measure of the performance in signal efficiency and background rejection. We also consider the background rejection at fixed efficiency points of $70\%$ and $85\%$, labeled by $R_{70}$ and $R_{85}$, respectively. Background rejection is defined as the reciprocal of the false positive rate at the specified true positive rate.

A summary of a variety of experiments is given in Table 1. The numbers in the table represent the median test score over 10 training runs, where the test score is always recorded at the epoch with the lowest validation loss. The quoted uncertainties for the rejections are the inter-quartile range.

## 5.2 Discussion

Table 1 shows that the family of models with only vector representations can indeed improve over the baseline, provided that we include at least the bilinear layer allowing the vector and scalar representations to mix. Moreover we find that adding the the $\mathrm{SO}(2)$ linear operations gives the vector network access to a minimal set of order-2 tensors, $R_{\hat{\mathbf{j}}}$, $\hat{\mathbf{j}}\hat{\mathbf{j}}^T$, and $I$ to enable it to exploit the axial symmetry of the data.

Table 1: Test metrics for training experiments on progressive model architectures. $R_{70}$ and $R_{85}$ indicate the test rejection at 70% and 85% signal efficiency, respectively. The percentage relative improvement in these metrics is also shown. Values shown are the median result over at least 10 training runs, per model type; errors quoted on rejection figures are the inter-quartile range.

| Model | $R_{70}$ | Impr.($R_{70}$) | $R_{85}$ | Impr.($R_{85}$) |
|---|---|---|---|---|
| Baseline (PFN) | $436 \pm 15$ | – | $112 \pm 3$ | – |
| Vector PFN | $1047 \pm 85$ | 140% | $235 \pm 12$ | 110% |
| **Tensor PFN** | $\mathbf{1176 \pm 103}$ | **170**% | $\mathbf{259 \pm 23}$ | **130**% |

In the case of the tensor family of models, there is a less substantial improvement when adding the $SO(2)$ linear layer. We expect that this is because the network with only bilinear operations is, at least in theory, able to learn the relevant operations on its own. Nonetheless, there is some improvement when adding this layer, so it would be reasonable to include both unless computational constraints are a concern.

Finally, we note that neither family of models performs even as well as the baseline, when no bilinear operations are allowed. This clearly demonstrates the effectiveness of a network which can mix $SO(3)$ representations.

# 6   Conclusion

In this work, we have introduced four modules of neural network architecture that allow for the preservation of $SO(3)$ symmetry. The Tensor Particle Flow Network (TPFN) shows promising results in our dataset, yielding up to $2.7\times$ improvement in background rejection, compared to the simple Particle Flow baseline model. We emphasize that the overall architecture of the PFN and TPFN are nearly identical; the improvement is entirely due to a drop-in replacement of standard neural layers with our covariant and bilinear layers. We also note that in our approach, the TPFN outputs a scalar which is rotationally invariant. However, it is also possible to obtain a covariant output by simply not apply the scalar pooling operations. This could be useful for many other applications, such as regression tasks, where covariant predictions are desired.

Moreover, we show that second-order tensor representations are required in order to exploit a locally-restricted class of equivariance with respect to the axial rotations $SO(2)_{\hat{j}}$. When computational constraints are a concern, it is possible to recover most of the performance of the Bilinear Tensor Network, by restricting it to a faster Bilinear Vector Network with the appropriate $SO(2)$ equivariant linear layer.

While the example application demonstrated here is of particular interest to the field of HEP, we expect our method can have great impact in other ares where vector-valued point cloud data is used. Finally, we note that while we demonstrated the modular elements of the TBN on a simple Deep Sets / PFN type network, it should also be possible to use these modules for creating equivariant Graph and attention based networks.

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
