# OpenReview forum: "Rethinking SO(3)-equivariance with Bilinear Tensor Networks"
_NeurIPS.cc/2023/Conference — Submitted to NeurIPS 2023_

### Official Review · Reviewer_p1Sx · 2023-06-10

**Soundness:** 2 fair
**Presentation:** 3 good
**Contribution:** 3 good
**Rating:** 6
**Confidence:** 5

**Summary:**

The paper proposes n permutation and SO(3) equivariant network for b-tagging in high energy physics. Permutation equivariance is required since the input data is a _set_ of track particle features, while SO(3) equivariance is required since these features are geometrically scalars or vectors whose rotational transformation law should be respected. The basic architecture is a particle flow network (PFN), which is based on deep sets. It consists of 1) a first SO(3)-equivariant subnetwork, applied to each track particle individually, 2) permutation invariant pooling via summation, 3) a second SO(3)-equivariant subnetwork, and 4) a final layer which extracts scalars. Internally, the network operates on scalars, vectors, and Cartesian 2-tensors. The first two are irreps of order $0$ and $1$, respectively, the third one is reducible (it would in principle decompose according to $1\otimes 1 \cong 0\oplus 1\oplus 2$).

The equivariant subnetworks employ a range of different SO(3)-equivariant mappings:
- Affine layers, i.e. linear maps followed by bias summation. The linear maps are made equivariant by broadcasting weights over the full representation dimension. Biases are only summed to scalars and 2-tensors since equivariant bias summation is impossible for vectors.
- Linear layers that are applied in local frames that are aligned along each individual jet's momentum axis $j$. This alignment is only specified up to rotations around the momentum axis, which is addressed by making the layers $\mathrm{SO}(2)\_j$ _gauge_-equivariant (the subscript labels the axis along which the subgroup is taken). Technically, the network seems to operate on restricted representations $\mathrm{Res}^{\mathrm{SO}(3)}_{\mathrm{SO}(2)} \rho$. Since the momentum axes $j$ are moving along with SO(3) rotations, the operation is as a whole still SO(3)-equivariant. Two explicit constructions of such SO(2) equivariant maps for vectors and 2-tensors are proposed. Due to the restricted equivariance requirement, these maps are less constrained than fully SO(3)-equivariant layers.
- Nine different _bilinear_ maps which map pairs of features of different types again to scalars, vectors and 2-tensors. This step allows (in contrast to the others) to transition between different representations.
- SO(3)-equivariant nonlinearities. For scalars, the authors use conventional ReLUs, while the nonlinearities for vectors and 2-tensors are acting on their norm (a common approach).

There is a single experiment, in which the models are trained as binary classifiers for b-tagging. The full equivariant model improves significantly upon a non-equivariant baseline and a version which ablates 2-tensor features.

**Strengths:**

The paper is well written and generally easy to follow. While I am not familiar with the b-tagging task and competing approaches, the empirical improvements over the baseline model seem very significant. Another strength is the use of bilinear mappings - most equivariant networks utilize only linear maps.

I really liked the idea of using locally $SO(2)\_j$ gauge equivariant operations besides fully SO(3) equivariant layers. The authors identified the additional geometric structure given by the momentum axes $j$ and addressed it appropriately.

**Weaknesses:**

My main concern with the approach is that the linear layers are not shown to be _complete_: in principle one could derive a basis of the most general equivariant linear maps (intertwiners) between the given representations. The authors show only the sufficiency of their layers regarding equivariance, but not the necessity. Indeed, I believe that quite some maps are in fact not the most general ones
(more details listed below).

To address this issue it is most convenient to work in the basis of irreducible representations, which the authors consciously avoid. For the following comments, note that scalars and vectors are irreps of order $0$ and $1$, while Cartesian 2-tensors are an irrep tensor product which decomposes according to $1\otimes 1 \cong 0\oplus 1\oplus 2$. Furthermore, intertwiners exist by Schur's lemma only between irreps of the same order, and are for SO(3) scaled identity matrices $\lambda\mathbb{I}$. For all non-trivial SO(2) irreps over $\mathbb{R}$ the spaces of irreps are 2-dimensional and are spanned by the irrep-endomorphisms ((1,0),(0,1)) and ((0,-1),(1,0)).

- The intertwiners for scalars and vectors (presented as "broadcasting") are complete since these are irreps. However, the broadcasting for $1\otimes 1$ tensors is overly restrictive, and there are actually three parameters, one for each irrep in the 2-tensor. The claim that the intertwiner space for 2-tensors is one-dimensional is repeated in line 172.
- The solutions for biases are indeed complete: biases can only be summed to trivial irreps, which exist with multiplicity 1 in scalars and 2-tensors, but not in vectors.
- It is furthermore possible to have intertwiners between scalars or vectors and 2-tensors, since the latter contain irrep orders 0 and 1 as subspaces (trace and antisymmetric part). These solutions are not used.
- The SO(2)-intertwiners between SO(3)-vectors in 3.1.1 seem complete, however, this is not proven but just claimed ("The set of all SO(2)-equivariant maps is _exactly_ ... [proposed parametrization]"). One can easily prove the completeness by observing that order 1 SO(3)-irreps decompose under restriction into the direct sum of an order zero and order 1 SO(2)-irrep, whose intertwiner spaces are 1 and 2-dimensional, respectively. The proposed parametrization has the same dimensionality.
- The SO(2)-intertwiners between Cartesian SO(3) 2-tensors are again not proven to be complete ("[equivariance] is satisfied when ... [proposed parametrization]" just claims sufficiency). Going to the irrep basis shows again that there are more possible solutions.
- In the case of bilinear maps, there are again some missing operations, e.g. combinations of two 2-tensors that result in a vector. All possible solutions follow directly from Clebsch-Gordan decompositions, which are well known for SO(3).
- An alternative approach to TReLU would be to apply three independent nonlinearities to the irreps contained in the 2-tensor. The equivariance of TReLU is not explicitly shown (this might be quite trivial to show).

As mentioned above, addressing these concerns would be easiest by switching to the irrep basis. As this would require a major revision, I am not sure whether this is the right way forward for this submission, or should rather be addressed in follow-up work. An alternative way to address these concerns would be to explicitly discuss completeness of intertwiner bases in general and prove it for each operation in which it holds. The equivariance of operations like TReLU or Eq. (4) should also be proven.

Another issue is that it is not well explained how the overall network remains SO(3)-equivariant despite intermediate operations only being SO(2)-equivariant. This is one of the most interesting aspects of the paper and deserves more attention.

It should also be discussed how this relates to other _gauge equivariant_ / _coordinate independent_ networks. The alignment of frames along the $j$-axis with remaining SO(2) ambiguity seems very similar to the SO(2)-structure (SO(2)-bundle of frames) considered by (Weiler et al. 2021), specifically their figure 53 (right).

The transition from SO(3) to SO(2) features is currently not sufficiently explained. I believe that the authors assume the restriction function $\mathrm{Res}^{\mathrm{SO}(3)}\_{\mathrm{SO}(2)}$ - please clarify this!

The group actions and the domains and codomains of A and L in sections 3.1.1 and 3.1.2 are nowhere defined. One can read them off between the lines, but they should be stated more clearly.

I am somewhat worried about the extent of the experiments. It would be nice to have a more thorough empirical analysis, for instance giving more ablations, showing training curves, investigating whether the equivariance is exact or due to numerical errors only approximate. Not all ablations discussed in the text of section 5.2 are shown in the table.

Finally, I wondered about the input features of the baseline, are they the same as for vector and tensor PFN? There should really be two baselines, with either set of features, to clarify that the improvement is really coming from the architectural changes instead of the input alone.
How exactly are the different models made comparable? Do they have the same number of parameters or the same computational cost to ensure a fair comparison?



**Questions:**

Most of my questions and suggestions are given in the "weaknesses" section. In addition I wondered about the following points:
- How can the SO(2)-equivariant layers be applied to the summed features, i.e. in the subnetwork $F$? As I understand, each jet comes with its own momentum axis $j$, but which axis is used after the summation?
- Why are the 2F features in the bilinear layers split in two groups? Other models consider all possible pairings of features (which scales quadratically instead of linearly in F).
- Is there a specific reasoning behind the norm-capping of VReLU and TReLU? Prior work like (Worrall et al., 2017) or (Weiler&Cesa, 2019) used ReLU-based "norm-nonlinearities" which do not saturate - this seems more intuitive and closer to the usual behavior of ReLUs to me.
- In how far is the saturation of VReLU and TReLU required to stabilize the training? Could you analyze or substantiate this claim with experiments?

**Limitations:**

The main limitation is in my opinion that the solutions are incomplete - which should be addressed more clearly. Other limitations, like the limitations of the bilinear ops due to not using Clebsch-Gordan decompositions or the lack of investigating the space of tensor nonlinearities are  adequately addressed.

Negative societal impacts are not to be expected.

---

> ### Author Rebuttal · Authors · 2023-08-10
>
> We thank the reviewer for a thoughtful response to our paper. We address the reviewer's specific questions below:
>
>  * "How can the SO(2)-equivariant layers be applied..."
>
> They can be used in exactly the same way. Because each parallel result of the subnetwork $\Phi$ is a proper vector quantity, so is their sum. The resulting sum can be passed into further $SO(2)_j$ layers, where $j$ could either be the original jet axis, which is common to all of the particles in the sum (as is the case in our implementation), or one could parameterize the layer with any other vector-valued $j$.
>
>  * "Why are the 2F features..."
>
> This is an arbitrary choice made purely for simplicity and to enable faster-running experiments.
> Still, with this simplified approach we find impressive an improvement over the baseline, which serves to highlight the main point of the paper, which is to demonstrate the benefit of the equivariant architecture.
> It would of course be possible (and presumably more expressive) to include all pairs of combinations, or to even use an attention mechanism to construct pairs.
> We hint at this possiblity in the conclusion, and leave for future work the integration of the modular, equivariant elements presented here into more sophisticated models such as transformers and GNNs.
>
>  * "Is there a specific reasoning behind norm-capping..."
>  * "In how far is the saturation of VReLU and TReLU..."
> These questions are related as indeed the reasoning for saturating activations was to prevent exploding magnitudes. We do not have particular experiments to this effect, _per se_; generally we were just completely unable to train reasonably deep networks without the saturating activation (we experienced exploding gradients). There may be other solutions to this problem, but our activation functions seem to work well empirically.

---

> > ### Comment · Reviewer_p1Sx · 2023-08-11
> >
> > I thank the authors for clarifying some of the questions. As there is no reaction regarding my concerns and suggestions in the weakness section I am sticking with my original rating.

---

### Official Review · Reviewer_gJAE · 2023-06-29

**Soundness:** 3 good
**Presentation:** 3 good
**Contribution:** 2 fair
**Rating:** 5
**Confidence:** 2

**Summary:**

The paper presents a method to handle complex geometric structured data, specifically SO(3) representations, through SO(3)-equivariance and judicious symmetry breaking. The technique improves computational efficiency and enhances the performance of a network operating on these representations. When applied to b-tagging, a High Energy Physics classification problem, it yielded a 2.7x improvement in rejection score over conventional methods.

**Strengths:**

1. It makes sense in machine learning to explore more efficient representation for data with symmetric structures. It is particularly important in many scientific fields such as HEP and material.
2. Using the proposed method, it shows a significant performance improvement compared with the baseline method.

**Weaknesses:**

1. Some explanation in the paper is difficult to be followed by ML researchers. This work deeply involves the task of B-jet identification at LHC experiments, but I’m not sure if these are sufficiently interesting for the ML and AI community.
2. In the related works, the authors mentioned that there had been numerous prior works on SO(3) equivariant models, but in the experiment only PFN with simulated datasets is implemented for comparison.  The numerical evidence in this paper looks not sufficient.

**Questions:**

Some comments:
1. It would greatly enhance the clarity of the paper if the authors provided an initial explanation of the concept of SO(3) symmetry. While the authors may be well-versed in this notion, it is essential to remember that many readers within the machine learning community may not be familiar with it. By providing a concise and accessible explanation, the authors can bridge the knowledge gap and ensure that readers grasp the significance of SO(3) symmetry in the context of their work.

2. In the introduction section, the authors make a claim that the proposed method is simpler compared to existing approaches. To strengthen this claim, it would be beneficial for the authors to support it with numerical or theoretical evidence. By presenting concrete results or theoretical analyses, the authors can substantiate their assertion and enhance the persuasiveness of their argument. This evidence will enable readers, including myself, to be convinced of the method's simplicity and its potential advantages over existing techniques.

3. The design of the new layers appears to be somewhat ad-hoc. While the paper demonstrates the equivariance of these layers, it would be valuable to explore whether alternative designs can achieve the same goal and what differentiates them. By discussing potential alternative designs or comparing the proposed layers with existing approaches, the authors can provide a more comprehensive understanding of the design choices made. This analysis will strengthen the paper's contribution by highlighting the unique aspects of the proposed layers and providing insights into their advantages over other potential designs.

4. If the main contribution of the paper lies in the introduction of the new layers, it would be beneficial to provide additional numerical evidence using different datasets. This would further validate the effectiveness of the proposed layers across a range of scenarios and reinforce their potential applicability beyond specific domains. Alternatively, if the focus of the paper is primarily on the performance improvement in the context of the LHC experiment, it would be essential to demonstrate the significance and relevance of this task. By providing additional context, motivation, and potentially exploring the broader implications of the performance improvement, the paper can better engage readers and establish the attractiveness of the addressed problem.

---

> ### Author Response · Authors · 2023-08-10
> **late rebuttal**
>
> Apologies for the late response. We thank the reviewer for a thoughtful response to our paper. We address points below:
>
>  * Some explanation in the paper...
> The b-tagging problem is an example of a class of symmetry-related problems, which are of interest to a much wider audience. We expect our novel methodology for dealing with restricted equivariance could be adapted to other fields with similar symmetry considerations. Our method is also versatile in the sense that it can be extended to other architectures such as GNNs.
>
> To facilitate the physics-specific discription of this work, we have also attached a supplementary figure with an illustration of the b-jet coordinate system.
>
>  * In the related works...
> Indeed, while we acknowledge prior work on SO(3)-equivariant models, our method is developed specifically to handle cases where the global SO(3) symmetry is expected to be broken by the underlying physical/generative data process. We are not aware of existing work that handles this case.
>
> Our experiments use high-fidelity physics simulators that are standard in the field of HEP. This is the only way to train and evaluate ML models, as it is technically impossible to produce labelled real data. The baseline PFN (a.k.a. Deep Sets) is chosen as it is a standard benchmark model on these types of tasks in physics/ML literature. But more importantly, our equivariant models are elaborations of the basic PFN architecture, so using a "plain" PFN model as the baseline is a natural choice to demonstrate the gain in performance directly attributable to our proposed methods, rather than architectural differences in the models.
>
> In other words, the baseline can be seen as a most extreme "ablation" where all of the improvements we propose are omitted.
>
>  * It would greatly enhance the clarity ... explaination of... SO(3)
> While we would love to make the paper as self-contained as possible, this is a difficult balance to strike given the page limit. We do feel that in the ML literature especially, there is a large and well-established body of work on equivariance stretching back several years with highly sophisticated foundations in group theory. We therefore feel that for the intended audience, the concept of the SO(3) group should be fairly well known; however, we will try to include a brief overview of the SO(3) group in the introduction if possible.
>
>  * In the introduction ...
> Thank you for pointing out the possible ambiguity in this statement. Our intention here is to describe the technical complexity of actually describing/understanding/implementing the method is simple. For example, we feel most would agree that working with vector dot products, cross products, etc, is much more intuitive and straightforward compared to, say, Clebsh-Gordon theory. Therefore we don't expect to have numerical evidence for this claim! However, we will certainly think of a better way phrase this argument.
>
>  * The design ...
> This is an interesting comment, and upon re-reading, it seems a reasonable impression. We will try to address it here in some detail, but will also work on the text to make the following arguments more clear.
>
> Firstly, the SO(2) layers are not ad-hoc in that we specifically analyzed criteria for equivariance (i.e., it must commute with $SO(2)_j$ rotations) and then came up with the general case in Eq. 4. The bilinear operations are perhaps ad-hoc, in that they are specialized and reduced cases of Clebsh-Gordon products of SO(3)-irreps. However, our physical intuition (i.e. inductive bias) suggests that nearly anything interesting one might want to compute on the given data (comprised of vectors and scalars), should be expressible in terms of the bilinear operations chosen.
> As for VReLU and TRelu: for VReLU we state a theorem that there's only one general form, however, the actual form of the scalar function $f(|\vec{x}|)$ we used, i.e. saturating relu, is indeed ad hoc. We describe one of a few that we tried, which was determined to work well empirically. The TReLU similarly is totally ad hoc and we admit that directly in the paper (while also pointing out that a more general version would be a function of the three principal invariants of the tensor).
> Our primary motivation and novel contributions in this paper is in dealing with the reduced/broken symmetry in an effective way, and we feel that the overall method is demonstrated soundly, even if some of the detailed choices are somewhat arbitrary.
>
>  * If the main contribution ...
> We agree that it would be more interesting to perform experiments on additional datasets. The dataset chosen of course is the one which directly motivated the architecture in the first place, due to the specialized symmetry of the underlying generative process. We are open to suggestions of additional point-cloud datasets that have a similar symmetry feature, if the reviewer is aware of any, and would be happy to conduct additional experiments provided they can be done in reasonable time.

---

> > ### Comment · Reviewer_gJAE · 2023-08-16
> >
> > Thank the authors for the detailed response. I'm not an expert on this topic, so I cannot evaluate how much novel technical contribution is given in this work. Nevertheless, I like this paper because it is quite well-written (the background part has been promised to be improved). So I tend to give an acceptance score.

---

### Official Review · Reviewer_ybFn · 2023-07-04

**Soundness:** 3 good
**Presentation:** 2 fair
**Contribution:** 2 fair
**Rating:** 6
**Confidence:** 3

**Summary:**

The authors propose a SO(3)-equivariant network operating on scalars, vectors and 2-tensors. They identify the corresponding equivariant linear layers and come up with a mixing strategy to mix different representations. Furthermore, SO(2)-equivariant linear layers are proposed to allow the scenario that SO(3) symmetry breaks into a subgroup SO(2) axial symmetry along a specific axis $\hat{j}$.  Finally, they demonstrate how it is appllied to b-tagging in High Energy Physics (HEP), where the data is rotational symmetric w.r.t. jet axis $\hat{j}$.

**Strengths:**

- The authors provide a good and simple implementation of SO(3) equivariant networks on scalars, vectors and 2-tensors. The weight matrix is carefully designed to preserve the symmetry. The way to mix different representations is also intuitive and easy to understand.
- The discussion on axial symmetry is well motivated and easy to follow, and the analysis is solid and sound.

**Weaknesses:**

- [Novelty] The idea of tensor-product-based representations is not new [1]. The main method (except the SO(2) part) looks a simple variation and the technique involved is pretty standard. Add discussion and comparison with existing tensor-product-representation-based methods could make the work more solid.
- [Evaluation] To show the effectiveness of mixing and SO(2) linear layers, I think it is better to put more intermidiate results (e.g., w/ and w/o SO(2) linear layers) in the main table.
- [Minor issues]: Eq. (1, 2, 3) use Einstein summation without declaration, which may cause confusion to readers without physics background. Line 168 should be "isotropic linear neuron of Eq. (2)" instead of "Eq. (1)".

[1] Finkelshtein, Ben, et al. "A simple and universal rotation equivariant point-cloud network." _Topological, Algebraic and Geometric Learning Workshops 2022_. PMLR, 2022.


**Questions:**

- How to understand Line 54 "We show that this kind of equivariant neuron is generally only possible with the introduction of order-2 tensor representations"? For me it seems everything should work well even if we only use scalars and vectors (like VectorPFN).

**Limitations:**

Limitations are not included in the manuscript.

---

> ### Author Rebuttal · Authors · 2023-08-10
>
> We thank the reviewer for a thoughtful response to our paper. We address  points below:
>
>  * "The idea of tensor-product-based neurons is not new..."
> We thank the reviewer for brining this work (and others) to our attention, and would be add a brief discussion comparing and contrasting our method. The primary difference that we emphasize is the partial symmetry breaking aspect of our network's architecture.
>
>  * "To show the effectiveness of mixing..."
> Indeed, we had ablation studies to this effect in an earlier preprint, but they were removed late in the editorial process. We have attached a PDF with a supplemental results table showing various ablations for the vector network. Unfortunately due to time constraints, only one ablation is available for the tensor network, although we would be happy to provide complete results within a few days if the reviewer is interested in viewing them during the discussion period.
>
>  * "Einstein summation..."
> Thank you, we have clarified the use of the Einstein summation convention, and fixed the mislabelled eqution. We have also added a figure illustrating the physical geometry of the b-jet events.
>
>  * "How to understand Line 54..."
> Thank you for pointing this out, upon rereading we agree it is not very clear. This is a reference to the fact that the $SO(2)_j$-equivariant neurons introduced in Eq. (4) intrinsically require the construction of an order-2 tensor, namely $jj^T$.
>
> Briefly the argument is that starting with a vector-valued neuron of the form $\vec{y} = A \vec{x}$,
>
> However, following our discussion regarding novelty above, and a similar discussion with reviewer NLGv, we prefer to remove this statement from the introduction and instead emphasize and clarify the order-2 tensor properties of the $SO(2)_j$ neuron.

---

### Official Review · Reviewer_er1i · 2023-07-06

**Soundness:** 2 fair
**Presentation:** 2 fair
**Contribution:** 1 poor
**Rating:** 4
**Confidence:** 3

**Summary:**

This work presents a lightweight architecture based on scalars, vectors, and tensors for learning in 3D, subject to $SO(2)$-equivariance in a known direction that varies by sample. They are motivated by the jet-tagging problem in high energy physics, in which a given batch is equivariant with respect to a certain axis (which may vary between batches), and test their framework on this problem.

**Strengths:**

Restricting to cross products and matrix products is new relative to previous work, which tends to focus more on the expressivity via irreducible representations of higher orders. The jet HEP dataset is also not commonly used in similar papers, and could provide a useful dataset for future papers. The proposed operations do indeed seem to be equivariant, and they are written out explicitly.

**Weaknesses:**

1. The proposed architecture is a simple restriction of many other existing architectures. The “tensor bilinear layer”, is a special case of the more general CG product that is now standard practice in other architectures, and it is not clear what benefits it has over a more general CG architecture. The nonlinearity of scaling by the vector or tensor norm is also not new: it is subsumed by the nonlinearity of applying an arbitrary function to the norm, and then scaling the vector or tensor by this value, which similarly was widespread in foundational works such as Tensor Field Networks (Thomas et al 2018) and subsequent works.
2. It does not seem like this architecture is particularly expressive, due to the use of low order features and the simple nonlinearity. (Note that prior work on the universality of point cloud architectures, and equivariant architectures more generally, usually requires making statements about polynomial approximation, where higher order tensor products are required to approximate higher degree polynomials — see e.g. Lorentz nets, Bogatskiy et al 2020, or Dym et al 2020 on the universality of point cloud architectures. Therefore, it seems to me that these simpler layers will likely have worse approximation properties.)
3. Based on the paper’s description (but the authors can correct if this is not the case), the motivating problem is really SO(2)-equivariant about a known but sample-dependent axis. Therefore, the discussion of the SO(3) CG product and other SO(3) architectures is somewhat misleading/confusing. It is also not explained clearly how using this paper’s SO(2)-equivariant architecture compares to the standard approaches in HEP based on the “transverse and longitudinal projections”.
4. The baselines and experiments are not sufficiently developed. For example, the only baseline is a DeepSet-style permutation invariant architecture, presumably on the raw 3D coordinates. However, the authors should compare to approaches using the apparently more standard “transverse and longitudinal projections”, as well as to SO(2)- or SO(3)-equivariant baselines on the raw coordinates. The paper claims that using a physically intuitive restriction of the space of possible operations is beneficial, but does not demonstrate this with an ablation study or by using e.g. a full CG product.

**Questions:**

1. What exactly are the “transverse and longitudinal” projections that other methods employ on this data, and why do other methods use them? Is returning to the full 3D coordinates with SO(2)-equivariance likely to be more efficient than reducing to these coordinates in the first place, from a sample complexity perspective?
2. Line 113 says “these methods all essentially rely on reducing the problem to vectors in a 2-dimensional plane”. Can the authors elaborate on what this means? If these methods use the jet direction to canonicalize the coordinate system, for example, they could easily use frame averaging (Puny et al 2021) to obtain equivariant outputs — but I am not sure if this is what they are doing or not.
3. What’s the difference between this approach and using a standard method for SO(2)-equivariance, such as circular convolutions about the direction j?
4. Line 48 claims that the tensor bilinear layer consists of “commonly known and physically intuitive operations”. Can the authors elaborate on the problem-specific physical intuitions that come with these operations? Is there some intuitive benefit to using physically intuitive operations as opposed to a more general framework relying on the full CG product?
5. Does the HEP problem enjoy translational symmetry as well?

Suggestions:
1. A diagram of the experimental set-up and the individual particle directions would make the paper and its motivation much more clear.
2. Order 2 tensors can be defined somewhere early in the paper.
3. The abstract refers to this method as “judicious symmetry breaking”. I think this is a misleading phrase, when really it is just that the symmetry group is a subgroup of SO(3) (rather than all of SO(3)). In other literature, symmetry-breaking refers to situations more complex than simply having (consistency) a smaller group than SO(3); for example, addressing the ambiguity between 6s and 9s in the MNIST digit dataset under rotation.

**Limitations:**

There is no potential negative societal impact. The paper does not address a potential lack of expressivity of the architecture.

---

> ### Author Rebuttal · Authors · 2023-08-10
>
> We thank the reviewer for a thoughtful response to our paper. We address specific points below:
>
>  * The baselines and experiments are not sufficiently developed...
> This is a misunderstanding; the baseline model _does_ receive as input the standard projections z_0 and d_0, cf. L234, in addition to everything else the other models get. The equivariant models _do not_ receive z_0/d_0, as they do not belong to a proper tensor representation.
>
> Regarding ablations, we agree that it's better to show the results of these studies, and have attached an expanded table of results for a series of ablations for the vector network. Unfortunately due to time constraints, only one ablation is available for the tensor network, although we would be happy to provide complete results within a few days if the reviewer is interested in viewing them during the discussion period.
>
>  * "What exactly are the transverse and longitudinal projections..."
> Thank you for pointing out we did not include a formal definition of d0 and z0, we will add these.
> Briefly it is sufficient to know that they are defined in terms of somewhat-complicated expressions involving cross products, dot products, and scalar products of momentum and position vectors; this was the original ansatz to design a network that is equipped with exactly these products as "primitive" operations.
>
> As for why d0 and z0 are preferred, it seems to be historical. Physicists have been identifying b-jets decades before tenable multi-variate methods were available. Projecting 3D information into a 1D quantity makes things easier in a "cut-based" event selection.
> One of our motivations is to demonstrate to practitioners that full 3D information can be much more powerful.
>
>  * Line 113...
> This is a subtle claim and it is challenging to balance the level of detail. Briefly, the "2D method" is to consider polar coordinates $(\Delta \theta, \Delta \phi)$ of each particle's momentum direction relative to the jet axis, which is considered locally flat-enough to associate particles with points on a 2D plane.
> For methods that consider only the "flow" of particles through the detector, it's sufficient to attach features such as charge, energy, etc. to these points and treat it as a 2D point cloud.
>
> In the b-tagging problem, the challenge in treating it as a purely SO(2) problem on a 2D coordinate system (e.g. to do circular convolution) is that the most useful observable, the impact parameter, is a fundamentally 3-dimensional quantity, and has no faithful representation in 2D. I.e. the manner in which the impact vector transforms under SO(2) rotation depends on its 3D orientation w.r.t. the jet axis.
> This is precisely the issue that our paper seeks to address: we "embed" a SO(2)-equivariant network within a SO(3) equivariant network to increases expressivity by exploiting the SO(2) restriction as much as possible.
>
> We also note that these methods are not mutually exclusive! E.g. circular convolution could be effected by, sampling from $SO(2)_j$ and applying it to vector/tensor-valued representations at any point in the network to yield a convolutional signal.
>
>  * "Line 48..."
> As mentioned earlier, the original ansatz is due to the explicit formulation of the d0 and z0 observables, in terms of scalar, dot, and cross products of vectors. The extension to order-2 tensors was an afterthought, upon considering that certain terms in the vector model (specifically, the dyadic product $jj^T$) are themselves rank-2 tensors.
> While these operations and associated representations can be thought of as a special case of CG products and SO(3) irreps, geometric tensors (including scalars and vectors) have their own straightforward algebra that is well known and comparatively easy to work with.
> From a (classical) physics perspective, there are almost never circumstances in where representations other than scalars, vectors, and order-2 tensors are called for. Therefore, our assumption (i.e. inductive bias) is that anything reasonably interesting that the network would like to compute should be possible with these products and representations.
> CG theory of course famously arises in quantum mechanics, however, one can see this necessity as arising from non-commutative operators, indistinguishable particles, and other circumstances in the theoretical treatment that physicists knows as "First Quantization". In brief, compare the procedure for adding angular momenta in the classical case (a simple vector sum) versus the quantum case (a CG direct sum of tensor products); why use the latter when the former is expected to be sufficient?
>
>  * "Does the HEP problem enjoy translational symmetry?"
> A good question, but it does not! We are measuring momentum eigenstates which emanate from the collision point to a far away surface (the detector). Observations cannot meaningfully be translated in either momentum or position space. The appearance of a position vector in this problem at all is a largely the reason extra care has to be taken in dealing with 3D rotations (as discussed above). We can include a brief discussion of this point, as another reviewer asked the same question.
>
>  * "A diagram of the experimental setup..."
> Good suggestion; we've included a diagram in the attached supplemental PDF, and propose to add it as a new figure in the next revision.
>
>  * "The abstract refers to ... symmetry breaking..."
> We feel our notion of symmetry breaking is consistent with common usage in physics; however, we would be interested in discussing and considering more carefully how it could be (mis)construed in the ML literature.
> From our perspective, the SO(3) symmetry is formally broken by the jet axis which fixes a ``special'' direction in space. That is, for a fixed $j$, we add terms to Eq. 4 which do not commute with arbitrary rotations, but which DO commute with arbitrary rotations about $j$, which for any given jet is expected to be a nearly exact symmetry of the underlying (physical) generative process.

---

> > ### Comment · Reviewer_er1i · 2023-08-21
> > **Thank you for the response**
> >
> > First, I unfortunately cannot find the supplemental PDF that the authors said they uploaded; to be fair (in case it was a mistake or I am missing it), I will write this response assuming that the PDF contains exactly what the authors claimed.
> >
> > I thank the authors for their detailed responses to all authors, which clarified some points that were originally unclear to me. In fact, I would strongly recommend that the equivariance proof provided in the rebuttal to Reviewer NLGv be included in the main paper, rather than the appendix (as suggested by the authors); I think this is quite important to the reader to understand.
> >
> > Overall, the use of SO(2)-equivariance in an SO(3)-equivariant architecture has some novelty. However, my concerns regarding the expressiveness of the architecture and comparison to appropriate baselines remains. The authors claim that “anything reasonably interesting that the network would like to compute should be possible with these products and representations”, but this should ideally be supported by experiments. For these reasons, as well as what I found to be an unclear presentation, I still do not think this paper is ready for publication. However, I will at least upgrade my score to a “weak reject,” and encourage the authors to address the reviewers’ comments in the next revision and resubmit.

---

### Official Review · Reviewer_NLGv · 2023-07-27

**Soundness:** 2 fair
**Presentation:** 3 good
**Contribution:** 2 fair
**Rating:** 4
**Confidence:** 4

**Summary:**

Before starting, I must mention that I am not a physicist, and therefore, I have focused on the machine learning aspects of the paper.

### Summary

This paper proposes the use of SO(3) equivariant neural networks for B-tagging. The method proposed builds upon Deep Sets and provides a quite constrained formulation of SO(3) equivariance. The paper further indicates that breaking SO(3) equivariance locally can be used to get better representations for the task at hand, while maintaining global equivariance to that group.


**Strengths:**

The paper is presented very clearly, well-structured, and it is in general easy to follow and understand.

**Weaknesses:**

### Concerns
* My main concern is that, to the best of my knowledge, it is not possible to obtain a certain equivariance without all networks being equivariant to that group. In particular, I do not understand how global SO(3) equivariance can be locally broken into SO(2)_j equivariance while preserving global SO(3) equivariance. In the best of my understanding, as shown in roughly all the papers on group equivariance, in order to have a network be equivariant to a certain transformation, all layers need to respect that equivariance. I think that clarifying how this is possible is *crucial* for the paper. It is important to note that the paper simply states this and does not provide proofs or analyses regarding this statement.
* I am very concerned with regard to the expressivity of the proposed algorithm. For instance, it has been shown in several works that equivariance can be obtained in expressive ways that do not have such hard restrictions as having biases be equal to zero –note that several similar very constraining restrictions are also defined for each of the mappings in the networks–, e.g., E(n)-equivariant Steerable CNNs, among many others. From what I understand, this work builds mostly upon the Deep Sets literature. However, this is by far not the most general way to obtain equivariance to a certain symmetry. I believe that the authors should at least state this clearly in the paper.
* The paper performs multiple ablations that are not found in the submission other than by conclusive statements. For instance, in line 322, the authors state: “Finally, we note that neither family of models performs even as well as the baseline, when no bilinear operations are allowed”. I believe that clearly showing the results of these ablations will strengthen the contributions of the paper. In general, such conclusive statements in their own are often vague and not very informative.
* In the final part of the conclusion it is stated that “it should be also possible to use these models for creating equivariant Graph and attention based networks”. Given that these families of networks –as stated earlier– are much more general than Deep Sets for equivariant formulations, I believe that this statement is not really easy to accomplish in practice.


**Questions:**

### Comments
* Line 34 - 35. What about translations?

### References
This paper misses several references in several parts of the paper. For instance:
* The second paragraph should include citations.
* Line 52. There are several works that perform equivariance relaxations.
* This paper is very similar to recent papers on Clifford Algebras, e.g., Clifford Group Equivariant Neural Networks. A discussion wrt these methods should be included.
* This paper also seems very similar to Vector Neurons. A discussion wrt such methods should be included.
* I have seen several nonlinearity formulations that look very much like the VReLU / TReLU, e.g., in Tensor field networks, Clifford Group Equivariant Neural Networks, etc. Please cite and discuss how the proposed nonlinearities are better.

### Introducing concepts
This paper does not introduce several things properly. For instance:
- Equation 4 falls completely out of the blue. To me, it is not clearly how the authors have reasoned to arrive at this.
- It is never explained that \alpha and \beta are in Figure 1.
- Why is x in |x| not bold?


**Limitations:**

See previous responses.

### Conclusion
In conclusion, I believe that the paper needs some work before I am able to support acceptance. I believe that there are several factors that need to be clarified, .e.g, how to get SO(3) equivariance with only SO(2)_j layers, for this to be an strong submission.

---

> ### Author Rebuttal · Authors · 2023-08-10
>
> We thank the reviewer for a thoughtful response to our paper. We address specific points below:
>
>  * "My main concern is that..."
> We agree that this point is of central importance to the entire work.
> The short answer is that the network is, in fact, globally equivariant w.r.t. SO(3).
> The reason for this is that the $SO(2)_j$-equivariant layers are themselves parameterized by the directional vector $j$, which transforms as $j \rightarrow R j$ under the coordinate rotation $R^{-1}$ in SO(3), so that the _layer itself_ can be thought of as transforming under SO(3) in such a way as to ``cancel out'' the local symmetry breaking.
> In particular, it can be shown that the neural weights $A$ in equation (4) transform formally as a order-2 tensor: $A \rightarrow R A R^T$. Therefore, the vector layer activation $y = A x$ satisfies SO(3) equivariance by transforming as a vector: $Ax \rightarrow R A R^T R x = R A x = R y$.
> The case for the tensor layer is analogous.
>
> The sense in which SO(3) is broken is that for _fixed_ $j$, the $SO(2)_j$ layer proposed here includes additional terms that would otherwise break $SO(3)$ equivariance. Since the underlying physical symmetry in the data being considered is formally $SO(2)_j$, the vector $j$ is considered fixed. In other words, while the network possesses a global symmetry, the data need not. In our example, any individual datum breaks SO(3) down to $SO(2)_j$ by specifying a particular direction.
> We show that we can take advantage of this by adding more expressive terms to the network that would not otherwise be allowed.
>
> We thank the reviewer for underscoring the subtlety of this argument, and would be happy to include some of the clarifying discussion in a revised draft.
> Due to page limits, we would also propose to include a detailed equivariance proof outlined here as an appendix.
>
>  * "I am very concerned with regard to..."
> We certainly do not argue that the method proposed is a path to a ``most general'' equivariant model. For instance, we are aware of work by Weiler, Jenner, et al. that establish group-convolutional methods as general linear equivarant maps under many groups and conditions.
>
> The method proposed here is of course not convolutional at all; while G-convolutions may be formally general for a given form of equivarance, many useful architectures are not purely convolutional, even when equivariance is at play, for example, patch-based CNNs.
>
> Instead, our intent is to design generalized analogs to perceptron/dense/fully-connected neurons which respect the particular symmetry of interest.
> The use of a Deep Sets architecture is for simplicity, given the point-cloud nature of the dataset, but is ultimately irrelevant to concerns of equivariance.
> As stated in the conclusion, we anticipate that these neural layers can be used as elements of other, more sophisticated equivariant models. For example, the vector and tensor layers proposed here could be as for projection operations in attention-based models, or to build edge convolution networks for DGCNN, etc.
>
>  * "The paper performs multiple ablations..."
> We thank the reviewer for pointing out this inconsistency; this remark was from an earlier preprint version, and was meant to be removed.
> Due to limited time to run ablations (particularly for the tensor model, which is slower to train), we decided to omit late in the editorial process.
>
> However, as it may be of interest, we are attaching a supplemental figure with an extended table of results, showing the detailed ablation studies for the vector PFN model.
> Unfortunately, only one ablation study is available for the tensor model (and with reduced statistics), again due to time constraints. We agree that this extended table would be of general interest to readers, and propose to include it in the next revision of the paper. Moreover, we should be able to produce analogous ablation studies for the tensor models within a few days, if the reviewer would like to see them during the discussion period.
>
>  * "In the final part of the conclusion..."
> We certainly agree that these are not _as_ easy as the Deep Sets, hence our decision to relegate this application to future work.
>
>  * "What about translations?"
> In this paragraph we are speaking generally and will rephrase. However, in our application, translation is not a relevant symmetry. The reason is that we are measuring momentum eigenstates which emanate from a fixed collision point to a far away surface (the detector). Experimental observations cannot meaningfully be translated in either momentum nor position space.
>
>  * References
> We agree with the reviewer on many points, and will fill out additional references as suggested.
> Regarding clifford algebras, we are unfamiliar with the work but presume you are referring to "Clifford Group Equivariant Networks" arxiv:2305.11141, which appears to have been submitted to arXiv one day _after_ we submitted the present manuscript for your consideration :)
>
> Regarding activation functions, we agree that additional references are warranted. However we are in particular not aware of a saturating ReLU unit to deal with exploding magnitudes. If the reviewer has a particular reference to give, we would be grateful.
>
>  * Equation 4...
> This is a purely geometrical result: the dyadic product $jj^T$ is an operator that when applied to a vector essentially projects the component parallel to $j$, and $I-jj^T$ therefore projects the perpendicular component. I.e. $jj^T v$ can be rewritten as $(j \cdot v) j$.
> We can include this brief comment in the text when Eq 4 is introduced, unless the reviewer feels an appendix is merited.
>
>  * alpha and beta in figure 1
> This is an oversight, the caption should explain that $\alpha$ and $\beta$ are feature indicies.
>
>  * conclusion
> Again we agree that this point is of central interest to the overall work. If you found our discussion above adequately clarifies the issue, we would be happy to include it in a revision to the main body of text.

---

> > ### Comment · Reviewer_NLGv · 2023-08-18
> > **Response to rebuttal**
> >
> > Dear authors,
> >
> > Thank you very much for your response.
> >
> > **We thank the reviewer for underscoring the subtlety of this argument, and would be happy to include some of the clarifying discussion in a revised draft. Due to page limits, we would also propose to include a detailed equivariance proof outlined here as an appendix.**
> >
> > +-> Unfortunately, I still cannot see how this can be achieved without breaking the global SO(3) equivariance. Could you please post a proof here so that I can engage in discussions about this with the other reviewers / ACs during the discussion period?
> >
> > **The method proposed here is of course not convolutional at all; while G-convolutions may be formally general for a given form of equivarance, many useful architectures are not purely convolutional, even when equivariance is at play, for example, patch-based CNNs.**
> >
> > +-> I understand. I think that making  this clear in the paper would help readability. In the same vein, probably making statements about generality of the method more subtle could better define the scope of the paper.
> >
> > **We agree that this extended table would be of general interest to readers, and propose to include it in the next revision of the paper. Moreover, we should be able to produce analogous ablation studies for the tensor models within a few days, if the reviewer would like to see them during the discussion period.**
> >
> > +-> It is perhaps too late to add them during the rebuttal period. However, I believe that these results would be very interesting for the general audience. I encourage the authors to include these results even if it is for the camera-ready version of the paper.
> >
> > **Regarding clifford algebras, we are unfamiliar with the work but presume you are referring to "Clifford Group Equivariant Networks" arxiv:2305.11141...**
> >
> > +-> I see. Please ignore this coment :)
> >
> > **We can include this brief comment in the text when Eq 4 is introduced, unless the reviewer feels an appendix is merited.**
> >
> > +-> A brief comment would be sufficient. Thank you!
> >
> > ### Summary
> >
> > Altogether I am happy with the author's response. However, I don't feel comfortable to support acceptance before I understand how exactly it is possible to break equivariance locally but not globally. I encourage the reviewers to provide a detailed proof in this regard.
> >
> > Best regards,
> >
> > Reviewer NLGv

---

> > > ### Author Response · Authors · 2023-08-21
> > > **equivariance proof**
> > >
> > > Dear reviewer, unfortunately I am not able to directly attach a document; however, since you have asked specifically for this material I have prepared a draft appendix with a more detailed proof of the global equivariance property, including a hopefully helpful, albeit unpolished sketch to illustrate the local rotations.
> > >
> > > I have posted a picture of the (anonymized) draft here, and will contact the AC to provide a proper PDF if possible.
> > >
> > > https://imgur.com/a/r3DULDD
> > >
> > > Moreover, we had tried to attach a (partial) table of ablations at the time of the original rebuttal; however, due to some technical issues it was not included in the original replies. We have likewise attached the original supplementary material as a "picture" at the following URL:
> > >
> > > https://imgur.com/a/E97aILS
> > >
> > > Thank you!

---

### Decision · Program_Chairs · 2023-09-21

**Decision:**

Reject

**Comment:**

This paper proposes a simple equivariant architecture for b-tagging. In contrast to reviewer gJAE I consider this to be well within scope for NeurIPS. The aim of simplifying architectures is also commendable, but as several reviewers noted the proposed simplification limits the expressiveness of the network (indeed it appears to be a special case of a more general class of architectures). As such it is necessary to perform rigorous experiments comparing the simplified method to more powerful (and complex) methods for intrinsically important tasks such as b-tagging. Hence I would recommend the authors to run experiments to find out whether the simplified architecture can be shown to perform similarly or better than more complex ones, and if so, resubmit the work.